# Biopsychosocial approach to understanding predictors of depressive symptoms among men who have sex with men living with HIV in Selangor, Malaysia: A mixed methods study protocol

**Zul Aizat Mohamad Fisal** [1], **Rosliza Abdul Manaf** [2]*, **Ahmad Zaid Fattah Azman** [2], **Gurpreet Kaur Karpal Singh** [3]

1 Faculty of Medicine and Health Sciences, University Putra Malaysia, Seri Kembangan, Malaysia,
2 Department of Community Health Sciences, Faculty of Medicine and Health Sciences, University Putra Malaysia, Seri Kembangan, Malaysia, 3 Klang District Health Office, Klang, Selangor, Malaysia

* rosliza_abmanaf@upm.edu.my

## Abstract

### Background

Depression is the most common psychiatric disorder reported among patients living with Human Immunodeficiency Virus (HIV), resulting from the intricate combination of biological, psychological, and social factors. Biopsychosocial factors can significantly impact the psychological well-being of men who have sex with men (MSM) living with HIV through social stigma, access and compliance to care, economic insecurity, relationship difficulties, and risky behavior. Compared to MSM without HIV, MSM living with HIV were more likely to be depressed. Despite specific vulnerabilities and health needs, MSM living with HIV remain understudied and underserved in Malaysia owing to legal, ethical, and social challenges.

### Objective

This is merely a published protocol, not the findings of a future study. This study aims to determine and explain the predictors of depressive symptoms among MSM living with HIV. Specifically, this study wants to determine the association between depressive symptoms among MSM living with HIV and biological, psychosocial, and social factors. Finally, the mixed methods will answer to what extent the qualitative results confirm the quantitative results of the predictors of depressive symptoms among MSM living with HIV.

### Methods

The study has ethical approval from the Medical Research Ethics Committee (MREC) of the Ministry of Health (MOH) NMRR ID-21-02210-MIT. This study will apply an explanatory sequential mixed methods study design. It comprised two distinct phases: quantitative and qualitative study design for answering the research questions and hypothesis. This study

**Data Availability Statement:** No datasets were generated or analysed during the current study. All relevant data from this study will be made available upon study completion.

**Funding:** Part of the work in this study is funded by the Putra Grant from Universiti Putra Malaysia with Vot Number 9693900. The funders did not and will not have a role in study design, data collection and analysis, decision to publish, or preparation of the manuscript.

**Competing interests:** The authors have declared that no competing interests exist.

will randomly recruit 941 MSM living with HIV in the quantitative phase, and at least 20 MSM living with HIV purposively will be selected in the qualitative phase. The study will be conducted in ten public Primary Care Clinics in Selangor, Malaysia. A self-administered questionnaire will gather the MSM's background and social, psychological, and biological factors that could be associated with depressive symptoms. For the quantitative study, descriptive analysis and simple logistic regression will be used for data analysis. Then, variables with a P value < 0.25 will be included in multiple logistic regression to measure the predictors of depressive symptoms. In the qualitative data collection, in-depth interviews will be conducted among those with moderate to severe depressive symptoms from the quantitative phase. The thematic analysis will be used for data analysis in the qualitative phase. Integration occurs at study design, method level, and later during interpretation and report writing.

## Result

The quantitative phase was conducted between March 2022 to February 2023, while qualitative data collection is from March 2023 to April 2023, with baseline results anticipated in June 2023.

## Conclusion

In combination, qualitative and quantitative research provides a better understanding of depressive symptoms among MSM living with HIV. The result could guide us to provide a comprehensive mental healthcare program toward Ending the AIDS epidemic by 2030.

## Introduction

Globally, over 264 million individuals suffer from depression, which is characterized by constant sorrow and a loss of pleasure or interest in activities that were before satisfying or enjoyable [1]. Depressive symptoms are generally known by the SIGECAPS mnemonic: sleep disorders (increased or decreased); interest shortfall; guilt (worthlessness, hopelessness, regret); energy shortage; concentration deficiency; appetite disorder (decreased or increased); psychomotor sluggishness or agitation; and suicidality. The presence of four SIGECAPS symptoms plus depressed mood or anhedonia suggests depression, and further screening should be considered [2]. Depression is distinct from regular mood swings and quick emotional reactions to ordinary stresses. It can be a serious health issue, mainly when persistent and moderate or severe [3]. Among persons with Human Immunodeficiency Virus (HIV) infection, depression is the most commonly documented psychological condition [4]. In a systematic review and meta-analysis, the pooled prevalence of depression among men who have sex with men (MSM) worldwide was 35%. Further continent analysis showed that Asian MSM had the highest pooled prevalence of depression at 37% [5]. Compared to MSM without HIV, MSM living with HIV were more likely to be depressed, with 43% to 58% depression prevalence based on the previous systematic review and study [6, 7].

By 2030, MSM are projected to account for the largest share of prevalent HIV infections in Malaysia, according to Asian Epidemic Modelling (AEM) [8]. In Malaysia, there was an estimated 220,000 MSM population with 21.6% HIV prevalence, making it the highest among other key populations: female sex workers (6.3%), transgender women (10.5%), and people who inject drugs (13.5%) [9]. Regarding depression, to date, the study involving MSM in Malaysia with the highest number of participants was conducted involving 622 mixed MSM

with and without HIV status, where 60.8% of them had Center for Epidemiologic Studies Depression (CES-D) scores of ≥ 16, indicating major depression symptoms [10]. However, in this study, the Patient Health Questionnaire-9 (PHQ-9) is utilized since it is half the length of the CES-D, is readily administered and scored, and is increasingly used across various patient groups in research and clinical settings [11].

Depression among MSM has significant effects on public health. Depression affects physical, educational, social, economic, psychological, and short- and long-term health effects for MSM living with HIV [12]. The presence of depressive symptoms may reduce an individual's motivation to engage in self-care and thus increase the likelihood of risky behavior and suicide [13, 14]. Among MSM living with HIV that have compulsive sexual behavior, having sex can become an alternative strategy to cope with negative emotions such as loneliness, anxiety, and stress [15]. Regarding viral load suppression, depression is a risk factor for 'imperfect' antiretroviral therapy (ART) adherence among MSM living with HIV that could jeopardize the Treatment as Prevention (TasP) strategy [16]. More importantly, the co-occurrence of HIV and depression was related to adverse health outcomes, such as low quality of life and deterioration of disease conditions [17].

A complicated combination of biological, psychological, and social variables could cause depression [3]. Therefore, the biopsychosocial approach has been used to structure guidelines, is used clinically, and discusses person-centered care, which can improve patient outcomes [18, 19]. The factors that increase depression among MSM living with HIV include younger age, unemployment, smoking, externalized HIV stigma, not being on ART, lack of self-efficacy, and low social support [20]. In addition, MSM are prone to mental health problems owing to sexual minority stress, homophobia, persecution based on sexual orientation, and stigmas that prevent healthy behavior [21, 22].

Addressing the mental health concerns of individuals living with HIV is essential to providing quality HIV care [9]. In many low and middle-income countries, MSM have fewer resources and are more likely to be hidden and stigmatized, compounding HIV acquisition and transmission risks and limiting access to the most basic services [23]. Continuing research, additional resources, political will, policy change, institutional transformation, community participation, and strategic planning and programming will be required to combat the increasing prevalence of depression among MSM living with HIV.

Despite specific vulnerabilities and health needs, MSM living with HIV remain understudied and underserved in Malaysia owing to legal, ethical, and social challenges. Considering the negative effects of depression on health outcomes and well-being, the determinants of depressive symptoms need to be explored for this vulnerable population. Mental health information from MSM living with HIV can be an important data source to complement existing research on HIV and mental health. The information can aid medical professionals and researchers in understanding the mental health challenges faced by the MSM living with HIV. The relevant bodies can gain insights into mental health needs and develop more effective interventions to address those needs by collecting and analyzing mental health information from MSM living with HIV. Furthermore, the information can inform public health policies and programs and improve the quality of care for MSM living with HIV. Finally, medical professionals can use this information to identify and treat mental health problems in their patients, ultimately leading to better overall health outcomes to support ending the AIDS epidemic by 2030.

This study used a mixed methods design as it can help explore complex phenomena such as depressive symptoms among MSM living with HIV/AIDS. This type of study is intended to give a deeper insight into a topic or issue than qualitative or quantitative research alone [24]. Researchers can use all the data collection tools available rather than being restricted to those types typically associated with quantitative or qualitative research [25]. The study will use the

biopsychosocial model as the theoretical framework to guide quantitative and qualitative phases. This framework incorporates biological, psychological, and social aspects to explain the intricacies and evolution of human behavior [26]. It is supposed to obtain information from the MSM's perspectives and experiences and focus on individual determinants of behavior related to depressive symptoms. Therefore, the protocol was set up to understand the predictors of depressive symptoms among MSM living with HIV and to discuss its relevance due to the negative impacts of depressive symptoms.

### Hypothesis

Hypothesis 1: There is a significant association between social factors and depressive symptoms among MSM living with HIV.

Hypothesis 2: There is a significant association between psychological factors and depressive symptoms among MSM living with HIV.

Hypothesis 3: There is a significant association between biological factors and depressive symptoms among MSM living with HIV.

## Materials and methods

This study has ethical approval from the Medical Research Ethics Committee (MREC) of the Ministry of Health (MOH) NMRR ID-21-02210-MIT. All information obtained in this study is confidential under applicable laws and regulations. No MSM's personal information will be published when publishing or presenting the study results. Individuals involved in the study and participating medical clinics, qualified monitors, auditors, the sponsor or its affiliates, and governmental or regulatory authorities may inspect and request a copy of medical records, where appropriate and necessary. Permission from the Director-General of Health, Malaysia, or relevant authorities will be obtained before publication. As part of the rights of the MSM as study subjects, they are permitted access to personal data obtained concerning them [27].

### General objective

To determine and explain the predictors of depressive symptoms among MSM living with HIV in Selangor.

### Specific objectives for quantitative study

1. To determine the prevalence of depressive symptoms among the MSM living with HIV in Selangor.

2. To describe the distribution of biological, psychological, and social factors of depressive symptoms among MSM living with HIV in Selangor.

3. To determine the association between biological, psychosocial, and social factors and depressive symptoms among MSM living with HIV in Selangor.

### Specific objective for the qualitative study

To explain further how the predictors influence depressive symptoms among MSM living with HIV in Selangor.

### Specific objectives for the mixed methods study

To determine to what extent the qualitative results confirm the quantitative result of the predictors of depressive symptoms among MSM living with HIV in Selangor.

### Study design

This study will apply an explanatory sequential mixed methods study design. There will be two main phases of execution, namely quantitative and qualitative study design for answering the research questions and hypothesis. A mixed methods study design is a type of research defined as combining research design and philosophical orientation by the researcher. The core components include a vibrant data collection and analysis of both study designs in response to the research question and hypothesis, combining both findings and implementing comprehensive methods into the specific study design to provide a logical approach within the philosophy framework. The study begins with quantitative data collection and data analysis. In the second phase, qualitative data will be collected and evaluated to explain or expand upon the quantitative results gained in the first phase. Qualitative data collection will be an in-depth interview with the MSM with moderate to severe depressive symptoms from the phase one study. The predictors of depressive symptoms will be further explained by the qualitative design. Data will be mixed by integrating quantitative and qualitative analysis while examining the study's findings and reaching conclusions.

This study applies integration at all study stages, where it occurs during study design, methods level, and interpretation and reporting levels. Regarding study design, the researcher gathers and analyzes quantitative data first, then uses those results to guide the collection and analysis of qualitative data. Meanwhile, integration at the methods level occurs through connecting, building, and merging approaches. The connecting approach occurs when the interview MSM are selected from those who responded in the quantitative phase. Integration through building occurs when the results from the quantitative phase inform the data collection approach for the qualitative phase, where the selected MSM will be those with PHQ-9 scores $\geq 10$. Integration through data merging occurred in this study when parallel questions with the quantitative phase were used in the qualitative phase. The researchers also will merge the PHQ-9 scores and MSM interview data to understand better how qualitative results explain quantitative results. Finally, integration at the interpretation and reporting level occurs when the researchers combine the data through a joint display table. The table will have three columns: (1) Quantitative results; (2) Qualitative interview explaining quantitative results; and (3) How qualitative findings help to explain quantitative results.

Pragmatism is a choice paradigm in this sequential explanatory explanatory sequential mixed methods study. It combines objective and subjective evidence that makes this design suitable for explaining complicated issues in public health, such as depressive symptoms among the MSM population. Fig 1 shows the Conceptual framework of explanatory sequential mixed methods study design on predictors of depressive symptoms among MSM living with HIV. Phase 1 will be the quantitative data collection, followed by the interface point. The MSM with moderate to severe depressive symptoms will be identified. The study then enters phase two, where qualitative data collection using an in-depth interview will be carried out among the MSM with moderate to severe depressive symptoms.

### Settings

Both study phases will be conducted in ten Primary Care Clinics in Selangor with the highest number of MSM living with HIV registered. Selangor has been chosen as the study location as a previous study shows a high prevalence of depressive symptoms among MSM in Selangor

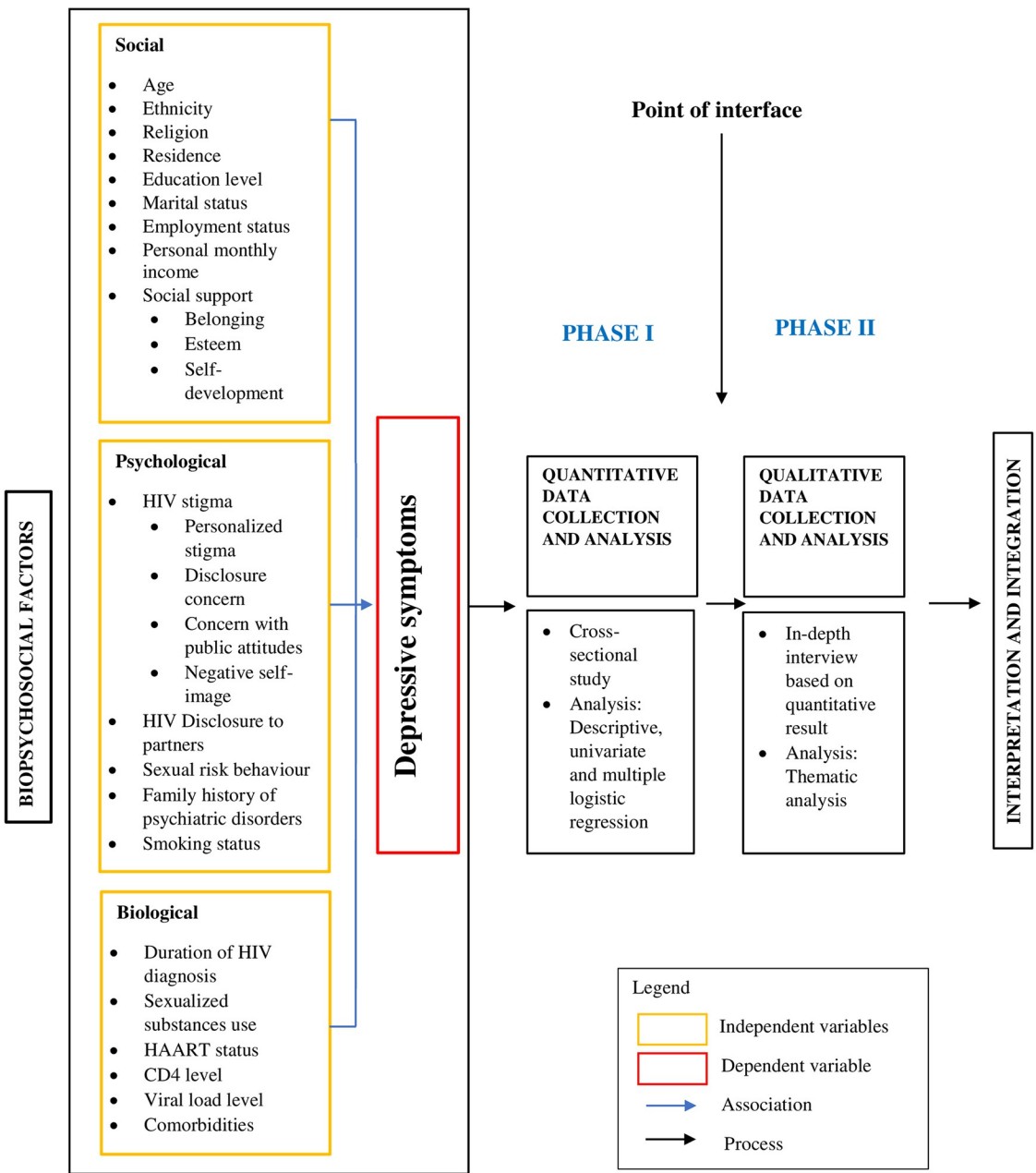

**Fig 1. Conceptual framework of explanatory sequential mixed methods study design on predictors of depressive symptoms among men who have sex with men (MSM) living with HIV.** Abbreviations: HIV, Human Immunodeficiency Virus; HAART, Highly Active Anti-Retroviral Therapy; CD4, Clusters of Differentiation 4.

(10). Quantitative data collection was conducted from March 2022 to February 2023, while qualitative data collection is from March 2023 to April 2023.

## Inclusion and exclusion criteria

The inclusion criteria were: (1) Malaysian; (2) able to read and understand the Malay or English language; (3) men self-identified as having sex with men based on the clinic database;

and (4) men with HIV diagnosis. The exclusion criteria were: (1) those aged less than 18 years old and (2) those known to have psychiatric disorders.

## Sampling methods

A proportionate random sampling was applied. The HIV databases from each clinic were obtained from the Selangor State Health Department to look for the list and the total number of patients in each clinic. Then, the required number of MSM in each clinic was calculated based on proportion to get 941 MSM. To build trust with the MSM and ease the data collection process, the researchers engaged with HIV clinic nurses and appointed them as research nurses. Next, the lists of MSM in each clinic were obtained to screen them for inclusion and exclusion criteria. Based on the screened name list of MSM in each clinic, random sampling is conducted to select the required number of MSM using a randomizer application. New master lists of selected MSM were extracted in an Excel sheet that contained their personal information, clinical information, and the clinic appointment date to remind the researcher to ask for written consent and to collect the data when the MSM visit the clinic. From the list, the researcher will create and insert a code to replace the MSM's names on the questionnaire and during the in-depth interview.

## Study instruments

A self-administered paper-based questionnaire will be used for this study to reduce non-response rates, as seen in another study [28]. A comprehensive translation will be done in Malay and English by those proficient in both languages. It will be distributed to the MSM in Malay or English based on their preference. The questionnaire will be divided into six sections, as follows:

i.  Section 1 for sociodemographic factors and clinical information
    Age, ethnicity, religion, residence, education level, marital status, employment status, monthly personal income, family history of psychiatric disorders, smoking status, CD4 level, viral load level, and other chronic diseases.

ii. Section 2 for depressive symptoms score
    Patient Health Questionnaire-9 (PHQ-9) was used to assess the severity of depressive symptoms, which includes nine items that focus on the Diagnostic and Statistical Manual of Mental Disorders, 4th edition (DSM-IV) for Major depressive disorder. The questionnaire measures the frequency with which each of the nine things bothered the individuals during the immediately preceding two weeks. Each item of PHQ-9 was scored on a scale of 0–3 (0 = not at all; 1 = several days; 2 = more than a week; 3 = nearly every day). The PHQ-9 total score ranges from 0 to 27 (scores of 0–4 as none; 5–9 as mild depression; 10–14 as moderate depression; 15–19 as moderately severe depression; and $\geq$ 20 as severe depression) [29].

iii. Section 3 for HIV-related stigma
    The 12-item short version of the HIV stigma scale was used to assess HIV-related stigma. The questions were further divided into four domains; personalized stigma, disclosure, negative self-image, and public attitudes. Responses were totaled to produce subscales ranging from 3 to 12 points; higher scores reflect a higher level of perceived HIV-related stigma [30].

iv. Section 4 for social support
    The Scale of Perceived Social Support in HIV (PSS-HIV) was selected. The scale can give

valuable data to establish strategies related to each person-specific level of needs and context and, thus, integrate into a practical tactic, a systemic hands-on approach for people living with HIV (PLHIV). The scale resembles the three top levels of Maslow's hierarchy of needs, composed of three sub-scales: (a) Belonging: the basic needs of support and safety; (b) Esteem: the acceptance, affection, and help from others; and (c) Self-development: the perception of achieving individual growth. The scale used 12 Likert-type items; all scored 1–5, with possible overall scores ranging from 12 to 60. Higher scores will indicate a higher Perceived Social Support [31].

v. Section 4 for high-risk behavior
The study will use the HIV sexual risk scale with nine items [32]. The questions will assess the history of high-risk sexual behavior in the past three months. All questions require the participant to answer "Yes" or "No.". If they answer 'yes' to any of the questions, they will be categorized as 'had recent high-risk behavior.'

vi. Section 5 for risk for sexualized substance use
Substance use questions will adopt the sociodemographic items from a study among MSM in Singapore by Tan et al. (2021) [33]. The questions will assess the history of sexualized alcohol use, sexualized drug use (amyl nitrite, methamphetamine, and Gamma-hydroxybutyrate/Gamma-butyrolactone), and ever-use erectile dysfunction drugs for sex. All questions require the participant to answer "Yes" or "No."

## Study processes

**a) Quantitative data collection.**   The researcher or research nurses will approach the MSM to participate in the study during clinic appointments or clinic visits. The research team had opportunities to recruit the MSM on different dates during doctor's follow-ups, pharmacist appointments, and blood-taking appointments. Research nurses are crucial in assisting research teams, ensuring that studies run efficiently and that MSM are safe and well-informed. They are eligible to recruit, gain consent, and collect data from the MSM. The researcher will explain the study protocol and obtain written consent from the MSM after a clinic visit in the consultation room or any available room in the clinic. Subsequently, the eligible MSM will be given a self-administered questionnaire to obtain the required information. After the MSM answer the questionnaire, their PHQ-9 score will be obtained on the spot. For ethical reasons, participants with moderate to severe depressive symptoms will be referred to Medical Officer at the respective clinic for further assessment.

## Quality control

**Face validity.**   Face validity concerns whether or not a test assesses what it is designed to measure. It will examine whether the questionnaire asks all the relevant questions and uses the appropriate language and language level [34]. The questionnaire will be given to three groups: supervisors, Family Medicine Specialists (FMS), and pilot MSM. Face validity will be conducted among a few MSM living with HIV in Bandar Botanik Health Clinic in Klang, Selangor. The feedback from the supervisors and the MSM upon the face validity will be assessed and amended.

**Content validity of the questionnaire.**   A forward and backward questionnaire translation was done according to the WHO translation guidelines. The English language expert personnel did the forward translation while the researcher did the backward translation. Then, the backward translation questionnaire was compared with the original questionnaire. After that, the questionnaire underwent content validity. The purpose of content validity was to look

for the magnitude in which the questionnaire's items thoroughly represented the construct of interest. For this study, the questionnaire was reviewed and discussed with two Public Health specialists, the Head of the HIV Unit and the Non-Communicable Disease (NCD) Unit from Selangor State Health Department and HIV Non-Governmental Organizations (NGO) representatives.

## Qualitative data collection method and process

**Interview guide for in-depth interviews.** The development of the interview guide for the in-depth interview is based on the biopsychosocial approach to help explain why the independent variables tested in the first phase were significant predictors of depressive symptoms. The interview protocol will next be tested on a subset of MSM from the pilot quantitative study with moderate to severe depression. This procedure will improve the quality of the questions through the MSM's input and the interviewer's familiarity with the questions.

**Field visit.** A field visit is one of the crucial steps in qualitative data collection. Before collecting the data, the researcher will develop a rapport with the health clinic staff and seek familiarity with the location. This step also allows the researcher to meet the health clinic's authority: the Family Medicine Specialist (FMS), the medical officer, the matron, sister in charge of the HIV clinic to obtain permission. A room or place will be designated to maintain privacy while collecting data. At that time, a brief presentation on the study procedure will be conducted to explain the data collection work process.

**Study sampling and procedure for qualitative phase.** Purposive sampling will be chosen in the qualitative phase. The MSM will be chosen from those with moderate to severe depressive symptoms or those with a PHQ-9 score of $\geq 10$ from the quantitative study. Sample size guidelines suggested an adequate range between 20 and 30 interviews [35]. Therefore, 20 MSM will be selected for the qualitative study. Suppose all or most MSM in the quantitative phase have moderate to severe depressive symptoms. Those with severe depressive symptoms will be chosen first, followed by moderately severe depressive symptoms and moderate depressive symptoms. The justification for this method is that the quantitative and qualitative data and their analysis might clarify and explain the statistical results through a more in-depth examination of the MSM's perspectives.

The study sampling is relatively small compared to the quantitative method, as it depends on the saturation point where no new information is acquired from the following interview. Data saturation is achieved when: (1) there is adequate information to replicate the study; (2) when the capacity to acquire new information has been attained; and (3) when further coding is no longer feasible [36].

Only the main researcher will conduct the in-depth interview for the qualitative study. The researcher will approach the selected MSM at the clinic during clinic visits. The researcher will explain the study protocol and obtain written consent from the participants after a clinic appointment in the consultation room or any available room in the clinic. The room used for the interview will be identified prior to the interview with permission from the clinic's head. Subsequently, an in-depth interview will be conducted to obtain the information required and will be voice recorded using two voice recorder devices. The interview will require MSM to tell their experiences regarding their depressive symptoms according to the biopsychosocial approach.

## Statistical methods and analysis

**Sample size calculation.** The sample was chosen based on a study by Perdue et al. (2003) [37], as this study gave the highest number of samples and the most feasible numbers to execute this research within the time frame.

P1 = 0.55 (Proportion of depression among MSM with education below college)
P2 = 0.45 (Proportion of depression among MSM with education level at college or higher)

$$n = \frac{\{1.96 \sqrt{2(0.55)(1-0.55)} + 0.62\sqrt{(0.55(1-0.55)) + 0.45(1-0.45)}\}2}{(0.55-0.45)2}$$

n = 392
Considering adjustment for comparison between 2 groups
= 392 x 2
= 784
Considering adjustment for non-response of 20%
= 784 + [(0.2) (784)] = 941

**Data analysis for quantitative study.** Data will be collected and analyzed using the IBM Statistical Package for Social Science (SPSS) version 25, involving descriptive and inferential statistics. The descriptive statistics will describe the MSM's characteristics: mean, standard deviation, mode, median, interquartile range, and percentage. Cross-tabulation and frequency counts will be used to assess the demographic items and the MSM's responses to survey scales. The data analysis utilizes both univariate and multivariate statistical methods. Simple logistic regression will be used to measure the associations between independent and dependent variables. A less rigorous threshold, such as a P-value < 0.25, should be applied to prevent omitting crucial variables from a model due to stochastic variation [38, 39]. Therefore, variables with a P value < 0.25 will be included in multiple logistic regression to measure the predictors of depressive symptoms. The binary outcomes of depressive symptoms will use a cut-off score ≥10. The scores of 0 to 9 represent none to subthreshold depression, and a score of 10 and above represents a patient's spectrum to possible major depression [40]. The level of significance at 0.05 and 95% confidence interval (CI) will be used to yield a significant result.

**Handling of missing data and outliers.** Missing data refers to the absence of expected data in a dataset that can occur for various reasons, including non-response and data entry mistakes. To assess the nature of missing data, it is necessary first to identify the missingness pattern: (1) systematic (obvious pattern) where missing values are only for the specific variable, or (2) random (no obvious pattern) and a small amount in a large data set. The amount of missing will then be determined by creating a variable group with missing and without missing variables and testing the difference (e.g., using the chi-square test). Missing data can be handled in three ways: 1) analyzes only data with complete values, 2) deletes cases or variables, and 3) data imputation. Analyzes only data with complete values can be done if missing is ignorable and the sample size is large enough. The options are to exclude cases list-wise or exclude cases pair-wise. If the researcher chooses to delete cases or variables, they must ensure it does not affect the entire result and achieve the study objective. The most likely way to handle missing data in this study is data imputation due to the large sample size, where missing data is estimated based on valid values of other variables or cases in the data set.

Any outliers in this study will not be removed first. The simple approach is to analyze with and without outliers and compare the results. If the results are similar, the outlier does not significantly influence the result. If the results change drastically, we will use the median (interquartile range) or 5% trimmed mean for descriptive results. The mean of the middle 90% of the observation will be taken by removing 5% of the upper and lower data. Data transformation will be carried out for the statistical test to avoid outliers affecting the result.

**Data analysis for qualitative study.** After each interview, the researchers will transcribe the audio recording verbatim. Next, a preliminary analysis will be conducted to identify codes and categories that may emerge from the data. There will be three coders during the process.

Multiple coders produce coding consistency or intercoder reliability by including two or more coders in the coding process, hence enhancing coding reliability [41]. This process will allow the researchers to get an overview of the data and become better interviewers by probing further in subsequent interviews and determining the information's depth and breadth. The interview document will be imported into NVivo Version 12 for analysis.

**Transcription.** Transcription is the first step of qualitative analysis, a tedious and meticulous process. It takes much time to complete the transcription of one voice recorder. Verbatim transcription is an ideal voice recording technique as it transcribes word by word, including the pause, mumbling, repetitive voice, or word. It helps the researcher feel the emotion and meaning of the discussion, which assists the researcher in further analysis [42–44]. In this study, verbatim transcription will be used in each interview session. Each MSM in the interview will use another name to protect their identity. The transcribed sheet needed to be re-read and checked against the voice recording to ensure no errors during the transcription process. Subsequently, each transcribed interview file will be saved using a MSM's code.

**Translation.** In the interview, most MSM will probably use the Malay language during the discussions, and maybe a few will use mixed language (Malay and English). The translation will only be done for coding quotes during the report-writing phase. It is a requirement for the university to do a final thesis in the English language. Therefore, a dual-language quote will be used to maintain the data's originality and make it more meaningful to the reader.

**Steps in qualitative data analysis.** Thematic analysis (TA) is one of the most widely used analyses. This analysis is preferred because of its flexibility, which helps the researcher identify the data set pattern permitted to answer the research question [44, 45]. In this study, a TA based on Braun & Clarke will be the method of choice to analyze the qualitative data. Table 1 shows the six steps in the TA [45].

**Ensuring rigor.** A qualitative study must ensure rigor and trustworthiness to ensure the data's validity and reliability. Trustworthiness will be explained using four concepts which are credibility (internal validity), transferability (external validity), dependability (reliability), and confirmability (objectivity) [44, 46].

**Credibility.** A credibility concept in a qualitative study is comparable with the internal validity of a quantitative study. This concept ensures the data's reliability, whether the interpretation of the informant's view is valid [35, 36]. The methods used include triangulation from several methods, member checks, reflexivity, and peer review [44, 46]. Credibility can be achieved using "member checks" to prevent misinterpretation or misunderstanding of the MSM's information. The transcription will be given back to the MSM for clarification. Any feedback is taken into consideration, and corrections will be made accordingly. Thus, the researcher's interpretations will align with the MSM's views. This method is also known as respondent validation [44].

**Table 1. The step of thematic analysis [45].**

| No | Steps | Description |
|----|-------|-------------|
| 1 | Data set familiarization | Involve verbatim transcription of voice recording. The researcher will repeatedly read the transcription for familiarization with the data pattern. |
| 2 | Initial coding | To organize the transcribed segment into meaningful coding (words or phrases). |
| 3 | Identifying themes | To merge the codes into themes with a broader meaning. |
| 4 | Reviewing themes | The coding within the themes will be reorganized and re-analyzing according to the research question. |
| 5 | Refining themes | To refine the theme to make it more explicit and specific. |
| 6 | Analysis report | Sufficient reporting with supporting data from the verbatim quote. |

This study will use peer review as a method for credibility. A discussion with the supervisory committee regarding the emerging raw data will be concluded for clarification. Recommendations and comments from the discussion will be considered for improvement. The researcher's reflexivity and positionality by making reflective notes during the data collection and analysis will also contribute to the information's credibility.

**Transferability.** Transferability reflects the general nature of the study. Transferability is established by presenting proof that the findings of the research study are applicable to others. The applicability can be achieved by a detailed description of the research process, such as research location, sampling method, and population sampling, as well as the study's outcome, together with evidence of the quotation from the MSM, including the field note [44, 46].

In this study, the qualitative study phase will be started after the data collection and the analysis of the quantitative phase. The MSM with moderate to severe depressive symptoms from the quantitative phase will be involved in the interviews. Hence, during phase one (quantitative) of data collection, a good rapport and healthy relationship will be established between the researcher and the MSM. After each interview, a qualitative study analysis will be conducted to look for any new information that can be included for further analysis. The data analysis result will be explained using the pre-determined themes from the predictors of the quantitative phase. It will be interconnected with subthemes, categories, and coding from the MSM's quotes.

**Dependability.** Dependability means the ability of the study to produce a similar result with a study's repetition. A study may experience difficulty yielding a consistent result because of a human factor in primary data collection. The reason might be the ever-changing human behavior depending on the surrounding environment [44, 46]. Therefore, these problems can be prevented by maintaining an adequate and detailed audit trail on the data collection method, analysis, and decisions on any inquiry.

The audit trail started with the ethics committee's ethical approval, followed by the data collection process, which includes field visits, sampling methods, and the execution of in-depth interviews. After that, the process continued with data management, analysis, and reporting. The final report will be discussed with the supervisory committee for feedback.

**Confirmability.** The other important concept of trustworthiness is confirmability. Confirmability is done by verifying the research findings with other modalities or the MSM. This study will verify the interview transcription with the voice recording to check the transcription accuracy. It will be performed every time after the completion of the verbatim transcription. Then, the preliminary analysis will be clarified.

## Safety considerations

**Confidentiality of quantitative study.** All information from this study will be kept and handled confidentially according to applicable laws and regulations. The personal identification Information will be replaced with research identification codes (ID Codes). Names and Identification Card (IC) numbers may not be incorporated into ID Codes. Access to master code lists or key codes is restricted. All data will be saved into a computer that is password protected. On completion of the study, data on the computer will be copied to a pen drive and erased. The pen drive and any hardcopy data will be stored in a locked cabinet belonging to the investigators and will be kept for a minimum of three years after completing the study. After three years, all the data will be destroyed. Electronic data files will be password-protected and encrypted. Files containing electronic data will be closed when computers are left unattended.

The consent form will be collected separately from the questionnaire. Only ID Codes will be used on the questionnaire. A master code list links personal identity information with ID Codes. The master code list will be kept in a locked cabinet separate from the survey form. Similarly, the consent forms will be housed separately from the questionnaire in closed cabinets or rooms. The MSM's identity will not be revealed when publishing or presenting the study, and they will be informed if new information relevant to consent becomes available.

**Confidentiality of qualitative study.**    Data from the quantitative study will be archived. The ID codes will also be used for each interview in the qualitative study. All the names of the MSM in the in-depth interview will be concealed, and no real name will be used during the interview. All data in both study designs will be entered into a computer that is password protected. On completion of the study, data on the computer will be copied to a pen drive and erased. The pen drive and any hardcopy data will be stored in a locked cabinet belonging to the researchers and will be kept for a minimum of three years after completing the study. After three years, all the data will be destroyed. The MSM will be informed if new information relevant to consent becomes available, and they may write to the researcher to request a summary of the completed study.

## Discussion

Phase I study used the cross-sectional design. Due to the simultaneous examination of both the outcome and the exposure, the temporal relationship between the two cannot be ascertained [47]. However, this design has its value where the point prevalence of depressive symptoms among a large sample of MSM can be very useful. Likewise, identifying protective and risk factors against depressive symptoms would benefit researchers and healthcare providers. The finding from this study could inform future interventions such as identifying at-risk individuals, developing tailored interventions, improving mental health outcomes by targeting predictors, and reducing HIV transmission by addressing depressive symptoms.

The research used a mixed methods design, integrating quantitative and qualitative data collection techniques. This method permits a more thorough understanding of the predictors of depressive symptoms among MSM living with HIV. The qualitative component of the study provides insights into the experiences of MSM living with HIV and how these experiences contribute to depressive symptoms. More importantly, integrating quantitative and qualitative data can significantly improve the value of mixed methods research [25]. Qualitative data can be used to assess the validity of quantitative results. In contrast, quantitative data can help to generate a qualitative sample or explain qualitative data findings [48].

Achieving meaningful integration is a key objective in mixed methods research. Integration strategies help researchers design their mixed methods studies with integration in mind and implement the integration plans during the conduct and reporting of the studies [49]. In sequential designs, the intent is to have one phase of the mixed methods study build on the other [50]. The integration during study design would provide a more profound and precious understanding of depressive symptoms and the related factors that are very sensitive to discuss outside the research context. At the methods level, integration can occur by linking data collection and analysis methods through connecting, building, merging, and embedding. In a single line of inquiry, integration may occur through one or more of these approaches [51]. In this study, integration in methods occurs at three levels when MSM from the quantitative phase are recruited in the qualitative phase (connecting), ensuring results from the qualitative phase built on quantitative findings (building) and combining the two datasets for analysis after independently studying each phase (merging).

Integration at the interpretation and reporting level can occur either through narrative, data transformation, or joint display [50]. Joint displays are increasingly acknowledged as highly effective techniques for achieving meaningful integration in mixed methods research [49]. Therefore, this study will use a joint display as it could identify new insights from the combined data and provide a broader and more comprehensive interpretation of MSM's responses and the research problem. By joint displays, researchers integrate the data by visually combining the quantitative and qualitative results to generate additional insights beyond the information received from the different outcomes. This can be accomplished by grouping the data in a figure, diagram, table, or graph [50]. By merging these two forms of data, researchers can acquire a deeper knowledge of the predictors of depressive symptoms among MSM living with HIV. This can aid in the creation of targeted mental health interventions and support services for this population.

No one shall be left behind according to the Sustainable Development Goals (SDGs) for Acquired Immunodeficiency Syndrome (AIDS) response [52]. The future of HIV response is intrinsically linked to global efforts to prevent non-communicable diseases (NCDs), including mental illness, to support ending the AIDS epidemic by 2030 [53]. We will work with the Ministry of Health (MOH) and Universiti Putra Malaysia (UPM) to develop effective dissemination strategies. The UPM has vast expertise in disseminating research materials and conclusions and summarising and revising the data to make it accessible to non-governmental organizations (NGOs) and other audiences. Individual consultations, group training, conference workshops, newsletters, journals, newspapers, fact sheets, and the website are creative means of disseminating information that the MOH and UPM will employ.

## Criteria for suspending or terminating the study

The study will be suspended or terminated by the sponsor or researcher's decision if safety concerns arise and the inability to sustain or further manage the study.

## Supporting information

**S1 File.**
(DOCX)

## Acknowledgments

The research can come with significant personal risks. Thanks to UPM for assisting in conducting and funding the study, UPM lecturers, and academic editors for reviewing and commenting on the protocol. We thank the Ministry of Health Malaysia and the Selangor Health State Department, who permitted us to conduct the research. We thank all Family Medicine Specialists, health care workers, and NGOs in public Primary Care Clinics (Bandar Botanik, Kapar, Meru, Kelana Jaya, Section 19, Section 7, Taman Medan, Seri Kembangan, Sungai Buloh, Puchong, Batu 9, Ampang, Kajang) for permitting to conduct the research in their respective workplace.

## Author Contributions

**Conceptualization:** Zul Aizat Mohamad Fisal, Rosliza Abdul Manaf, Gurpreet Kaur Karpal Singh.

**Formal analysis:** Zul Aizat Mohamad Fisal, Rosliza Abdul Manaf, Ahmad Zaid Fattah Azman, Gurpreet Kaur Karpal Singh.

**Methodology:** Zul Aizat Mohamad Fisal, Rosliza Abdul Manaf, Ahmad Zaid Fattah Azman, Gurpreet Kaur Karpal Singh.

**Supervision:** Rosliza Abdul Manaf, Ahmad Zaid Fattah Azman, Gurpreet Kaur Karpal Singh.

**Writing – original draft:** Zul Aizat Mohamad Fisal.

**Writing – review & editing:** Zul Aizat Mohamad Fisal, Rosliza Abdul Manaf, Ahmad Zaid Fattah Azman, Gurpreet Kaur Karpal Singh.

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
