## [Decision Letter · Decision Letter 0]

22 Dec 2022

PONE-D-22-23739Biopsychosocial approach to understanding predictors of depressive symptoms among men who have sex with men living with HIV in Selangor, Malaysia: A mixed-methods study protocolPLOS ONE

Dear Dr. Abdul Manaf,

Thank you for submitting your manuscript to PLOS ONE. After careful consideration, we feel that it has merit but does not fully meet PLOS ONE’s publication criteria as it currently stands. Therefore, we invite you to submit a revised version of the manuscript that addresses the points raised during the review process.

Please note that we have only been able to secure a single reviewer to assess your manuscript. We are issuing a decision on your manuscript at this point to prevent further delays in the evaluation of your manuscript. Please be aware that the editor who handles your revised manuscript might find it necessary to invite additional reviewers to assess this work once the revised manuscript is submitted. However, we will aim to proceed on the basis of this single review if possible. 

We look forward to receiving your revised manuscript.

Kind regards,

Jamie Males

Editorial Office

PLOS ONE

Journal Requirements:

Reviewers' comments:

Reviewer's Responses to Questions

**Comments to the Author**

1. Does the manuscript provide a valid rationale for the proposed study, with clearly identified and justified research questions?

Reviewer #1: Partly

2. Is the protocol technically sound and planned in a manner that will lead to a meaningful outcome and allow testing the stated hypotheses?

Reviewer #1: Partly

3. Is the methodology feasible and described in sufficient detail to allow the work to be replicable?

Reviewer #1: Yes

4. Have the authors described where all data underlying the findings will be made available when the study is complete?

Reviewer #1: Yes

5. Is the manuscript presented in an intelligible fashion and written in standard English?

Reviewer #1: Yes

6. Review Comments to the Author

You may also provide optional suggestions and comments to authors that they might find helpful in planning their study.

Reviewer #1: ABSTRACT

- Abstract should be clearer in objectives that this is only a published protocol and not the results from the study that will be conducted in the future

- Add “%” to depression prevalence estimates

BACKGROUND

- Is the prevalence for depression (43% t 51%) for a region, country or globally?

- Page 4, define ART as it appears for first time

- Replace HIV-positive individual with “individuals living with HIV”

- Recommend adding more background statistics like the size of the study population in Malaysia to demonstrate public health impact. Can also give the proportions of Malay MSM compared to other Malay people living with HIV.

- Is there any theoretical framework being used to generate the research questions and also to help guide instrument development for both phases of research?

METHODS

- Pg. 7, end of bullet 3, missing word “factors” at the end of the statement

- Last sentence on page seven written in past tense so there is tense discord

- Pg 8, should it be philosophy framework?

- Pg 8, what if all the sample or most of the sample in the quantitative phase has moderate or sever depression, how will the 20 participants for the qualitative phase be selected?

- Pg. 9/10 – are there eligibility criteria related to HIV status?

- Pg. 10, is the questionnaire computer administered or otherwise?

- Will an interview guide be developed for the in-depth interviews

- Logistic regression is proposed for analysis, are all the outcome variables dichotomous?

- Is there any plan for missing variables or outliers?

- Recommend having more than one coder for the qualitative data analysis

7. PLOS authors have the option to publish the peer review history of their article (what does this mean?). If published, this will include your full peer review and any attached files.

Reviewer #1: No

---

## [Author Response · Author response to Decision Letter 0]

2 Jan 2023

Dec 31, 2022 

Rebuttal Letter to manuscript:

Biopsychosocial approach to understanding predictors of depressive symptoms among men who have sex with men living with HIV in Selangor, Malaysia: A mixed-methods study protocol

Zul Aizat Mohamad Fisal1, Ahmad Zaid Fattah Azman2, Gurpreet Kaur3, Rosliza Abdul Manaf2* 

1Faculty of Medicine and Health Sciences, University Putra Malaysia 

2Department of Community Health Sciences, Faculty of Medicine and Health Sciences, University Putra Malaysia 

3Klang District Health Office, Selangor, Malaysia

Dear Academic Editor and reviewer,

Thank you for your comments and recommendations, which allowed us to improve the paper. In the updated manuscript, we expect to answer all the issues identified. In this document, we answer all the questions the reviewers highlighted.

Comments are shown in bold font, followed by our answer/comment in normal font. The corrections/changes in the manuscript are displayed through the track changes.

Reviewer #1: 

1. ABSTRACT

- Abstract should be clearer in objectives that this is only a published protocol and not the results from the study that will be conducted in the future

- Add “%” to depression prevalence estimates

Response:

-Dear reviewer, to clarify, the sentences were added as suggested: This is merely a published protocol, not the findings of a future study.

-% added to prevalence estimates.

2.BACKGROUND

- Is the prevalence for depression (43% t 51%) for a region, country or globally?

- Dear reviewer, the prevalence was based on the previous systematic review and study (sentences added).

- Page 4, define ART as it appears for first time

- Dear reviewer, ART defined as antiretroviral therapy (ART)

- Replace HIV-positive individual with “individuals living with HIV”

- Dear reviewer, the sentence was replaced as suggested.

- Recommend adding more background statistics like the size of the study population in Malaysia to demonstrate public health impact. Can also give the proportions of Malay MSM compared to other Malay people living with HIV.

- Dear reviewer, the prevalence of MSM population in Malaysia and MSM living with HIV compared to the others key populations has been added.

- Is there any theoretical framework being used to generate the research questions and also to help guide instrument development for both phases of research?

- Dear reviewer, the sentences added: “The study will use the biopsychosocial model as the theoretical framework to guide quantitative and qualitative phases. This framework incorporates biological, psychological, and social aspects to explain the intricacies and evolution of human behavior”.

3.METHODS

- Pg. 7, end of bullet 3, missing word “factors” at the end of the statement

- Word added.

- Last sentence on page seven written in past tense so there is tense discord

-The sentence was change to future tense.

- Pg 8, should it be philosophy framework?

-Yes, it should be philosophy framework.

- Pg 8, what if all the sample or most of the sample in the quantitative phase has moderate or severe depression, how will the 20 participants for the qualitative phase be selected?

- However, suppose all the respondents or most respondents in the quantitative phase have moderate or severe depression. In that case, those with severe depression will be chosen first, followed by moderately severe depression and moderate depression. The justification for this method is that the quantitative and qualitative data and their analysis might clarify and explain the statistical results through a more in-depth examination of the respondents’ perspectives.

- Pg. 9/10 – are there eligibility criteria related to HIV status?

- Yes, eligibility is MSM living with HIV. Sentences were corrected.

- Pg. 10, is the questionnaire computer administered or otherwise?

- Dear reviewer, a self-administered paper-based questionnaire will be used for this study to reduce non-response rates, as seen in another study.

- Will an interview guide be developed for the in-depth interviews

- Yes, the interview guide will be developed. The sentences were added under the Qualitative data collection method and process.

- Logistic regression is proposed for analysis, are all the outcome variables dichotomous?

- Dear reviewer , the binary outcomes of depressive symptoms will be using a cut-off score of 10. The scores of 0 to 9 represent none to subthreshold depression, and a score of 10 and above represent a spectrum of a patient to possible major depression based on depressive symptoms.

- Is there any plan for missing variables or outliers?

- Dear reviewer, all missing data will be assigned with missing value codes. The rules are: 1) missing value codes must be of the same data type as the data they present (e.g: for missing numeric data value codes must also be numeric; 2) Missing value codes cannot occur as data in the data set, and 3) By convention, the choice of the digit is -99.

-Any outliers will not be removed first. The simple approach is to analyze with outliers and without outliers and compare the results. If the results are similar, the outlier does not significantly influence the result. If the results change drastically, we will use the median (inter-quartile range) or 5% trimmed mean for descriptive results. The mean of the middle 90% of the observation will be taken by removing 5% of the upper and lower data. For the statistical test, we will carry out data transformation to avoid the result being affected by the outliers.

- Recommend having more than one coder for the qualitative data analysis

- Dear reviewer, thank you for the recommendation. Sentences added “ There will be three coders during the process. Multiple coders produce coding consistency or intercoder reliability by including two or more coders in the coding process, hence enhancing coding reliability”.

Dear Editors

We thank the reviewer for their generous comments on the manuscript and have edited it to address all the concerns.

We believe that the manuscript is now suitable for publication in Plos One.

Sincerely

Zul Aizat Mohamad Fisal

Doctor of Public Health candidate,

Faculty of Medicine & Health Sciences,

Universiti Putra Malaysia 

On behalf of all authors.

---

## [Decision Letter · Decision Letter 1]

12 Apr 2023

PONE-D-22-23739R1Biopsychosocial approach to understanding predictors of depressive symptoms among men who have sex with men living with HIV in Selangor, Malaysia: A mixed-methods study protocolPLOS ONE

Dear Dr. Abdul Manaf,

Thank you for submitting your manuscript to PLOS ONE. After careful consideration, we feel that it has merit but does not fully meet PLOS ONE’s publication criteria as it currently stands. Therefore, we invite you to submit a revised version of the manuscript that addresses the points raised during the review process.

Please take note that you need to address the reviewer's comments accordingly to strengthen the quality of your protocol. 

We look forward to receiving your revised manuscript.

Kind regards,

Rogie Royce Carandang, RPh, MPH, MSc, PhD

Academic Editor

PLOS ONE

Reviewers' comments:

Reviewer's Responses to Questions

**Comments to the Author**

1. Does the manuscript provide a valid rationale for the proposed study, with clearly identified and justified research questions?

Reviewer #2: Partly

2. Is the protocol technically sound and planned in a manner that will lead to a meaningful outcome and allow testing the stated hypotheses?

Reviewer #2: Partly

3. Is the methodology feasible and described in sufficient detail to allow the work to be replicable?

Reviewer #2: No

4. Have the authors described where all data underlying the findings will be made available when the study is complete?

Reviewer #2: No

5. Is the manuscript presented in an intelligible fashion and written in standard English?

Reviewer #2: Yes

6. Review Comments to the Author

You may also provide optional suggestions and comments to authors that they might find helpful in planning their study.

Reviewer #2: Overview

The authors present a robust study protocol of a mixed-methods study that follows an explanatory-sequential design to understand factors associated with depressive symptoms among Malaysian MSM living with HIV. I commend the researchers for dedicating their time and resources to an underservices and under-engaged population. Although an important topic, there are major issues with this manuscript concerning a) the framing of the study, b) the methods (e.g.,: sample size calculation, unclear inclusion criteria and setting description), and c) the discussion. Most importantly, attention is not paid to how the data will be integrated, and this is core to a mixed-methods study. I have outlined specific concerns below.

Abstract

1. The background could use further attention to the socioecological factors mentioned that affect people living with HIV. It isn’t enough to state that these factors exist – how do they related specifically to the psychological wellbeing of people living with HIV?

Introduction

1. The first paragraph would greatly benefit from varied sentence structure so that most sentences do not begin with “Depression”.

2. MSM should be defined prior to use of the abbreviation, as should HIV.

3. You should state the prevalence of depression among MSM to further make the point that depression prevalence is elevated among MSM with HIV.

4. “MSM is likely to be the main key population in Malaysia by 2030.” I don’t think this says what you mean for it to say. Do you mean that “By 2030, MSM are projected to account for the largest share of prevalent HIV infections in Malaysia”

5. You do not sufficiently set up how “mental health information from MSM living with HIV” will be used. What value does this information bring? Does it inform interventions?

6. Respectfully, I do not follow the decision to end the introduction by saying that this work will identify gaps in health services. You set up the protocol by wanting to understand factors that are associated with depression among MSM, and then discuss how this is of relevance because depression has been associated with non-adherence to HIV medications and other sexually risky behaviors. I would just make it clear that you’re focused on individual determinants of behavior – your data is not going to speak to gaps in health services because, from what I can tell, you are not proposing to assess these factors.

Methods

1. I am alarmed at the last sentence of the first paragraph in the methods: “The respondents will be given access to personal information… upon request”. Please clarify the meaning of this.

2. The authors explain the phases and rationale for such phases, of their mixed-method design, well. Yet no attention is paid to data integration.

3. I do not follow the sample size justification. A sample size of nearly 1000 MSM with HIV to recruit in a period of approximately 4 months, even across 10 sites with an established sampling frame, seems improbable. Might you be able to speak to the feasibility of this based on past work by your team and or additional information about the recruitment sites? I also do not understand why the sample size calculation is not grounded in your statistical approaches.

4. If you are interested in an outcome of depression, why are you not screening participants to enroll those with clinically significant depressive symptoms? There is no guarantee that you will result in a sample of MSM with HIV who also have depression that is large enough to facilitate your analysis

5. Inclusion criteria – “Men identified as homosexual based on clinic database”. This will exclude MSM who are not homosexual. MSM is a behavior, not an identity.

6. It seems that site selection is conflated with participant selection. For example – criterion 4 is that there must be more than 200 MSM with HIV. This isn’t a participant level factor.

7. No where does it say that the participants must have HIV in the inclusion criteria

8. Section 4 social support. PLHIV is used and not defined

9. Might you consider measuring other likely relevant constructs including: self-efficacy in adhering to medications, internalized MSM related stigma

10. It is not clear how random sampling will occur.

11. It is not clear why you would need to do a pilot study with 30 people to look for internal consistency of the select items. Have these measures not performed well in other samples? I could see you wanting to pilot if you were creating original measures.

12. The section on the ethical reasons of linking people who screen with depression to care should be included prior to the section on analysis.

13. You have not written about exploring the nature of the missing data nor considered imputation to address missingness, which you could do with a sample size as large as you have.

14. With a proposed sample size as large as you have and for a research question like you pose, it is a missed opportunity to not engage in structural equation modeling. You’re really interested in a lot of latent variables that are challenging to conceptualize and measure that affect the experience of depression in an individual who is MSM and also HIV+

15. Despite this being a mixed methods study, no attention is paid to the integration of the data.

Discussion

1. I find it problematic to describe recall measures as buffering against the limitation of cross-sectional data. The data is still cross-sectional in nature. Perhaps instead of stating this, you could reiterate the value of the cross-sectional data (point prevalence of depression among a large sample of MSM is very useful, understanding of factors associated with depression informs intervention, or identifying factors that are protective against depression is useful too!)

2. Overall – it is less important to be discussing how withdrawal of participants will be handled. I would save that for unpublished material. What would be most informative is a discussion of how this data will be integrated (quant/qual mixing), reiteration of the relevance of this work. ETC.

7. PLOS authors have the option to publish the peer review history of their article (what does this mean?). If published, this will include your full peer review and any attached files.

Reviewer #2: No

---

## [Author Response · Author response to Decision Letter 1]

20 Apr 2023

Dear Academic Editor and reviewers,

Thank you for your comments and recommendations, which allowed us to improve the paper. In the updated manuscript, we have tried to improve all the issues identified. In this document, we answer all the questions the reviewers highlighted.

Reviewers comments are shown in bold font, followed by our answer/comment in normal font. The corrections/changes in the manuscript are displayed through the tracked changes.

Reviewer #2: 

Abstract

1. The background could use further attention to the socioecological factors mentioned that affect people living with HIV. It isn’t enough to state that these factors exist – how do they related specifically to the psychological wellbeing of people living with HIV?

Thank you for the comments. We have corrected the background, but instead of using the word ‘socioecological factors’, we use biopsychosocial factors to link it with the title and the whole protocol.

Introduction

1. The first paragraph would greatly benefit from varied sentence structure so that most sentences do not begin with “Depression”.

The sentence structure has been rephrased and modified to ensure fluency and clarity.

2. MSM should be defined prior to use of the abbreviation, as should HIV.

The abbreviations have been defined.

3. You should state the prevalence of depression among MSM to further make the point that depression prevalence is elevated among MSM with HIV.

The prevalence of MSM has been stated, followed by the prevalence of MSM living with HIV for comparison.

4. “MSM is likely to be the main key population in Malaysia by 2030.” I don’t think this says what you mean for it to say. Do you mean that “By 2030, MSM are projected to account for the largest share of prevalent HIV infections in Malaysia”

Thank you for the suggestion. The sentence has been corrected as suggested.

5. You do not sufficiently set up how “mental health information from MSM living with HIV” will be used. What value does this information bring? Does it inform interventions?

The paragraph has been corrected with an addition on the value of mental health information in paragraph 6.

6. Respectfully, I do not follow the decision to end the introduction by saying that this work will identify gaps in health services. You set up the protocol by wanting to understand factors that are associated with depression among MSM, and then discuss how this is of relevance because depression has been associated with non-adherence to HIV medications and other sexually risky behaviors. I would just make it clear that you’re focused on individual determinants of behavior – your data is not going to speak to gaps in health services because, from what I can tell, you are not proposing to assess these factors.

The sentence on gaps in health services has been removed and replaced with the suggested correction.

Methods

1. I am alarmed at the last sentence of the first paragraph in the methods: “The respondents will be given access to personal information… upon request”. Please clarify the meaning of this.

The sentence has been changed and with reference for clarity. New sentence: “As part of the rights of the respondents, they are permitted access to personal data obtained concerning them.”

2. The authors explain the phases and rationale for such phases, of their mixed-method design, well. Yet no attention is paid to data integration.

A paragraph on data integration has been added in the methodology section. The part on data integration in detail was added in the discussion section as well.

3. I do not follow the sample size justification. A sample size of nearly 1000 MSM with HIV to recruit in a period of approximately 4 months, even across 10 sites with an established sampling frame, seems improbable. Might you be able to speak to the feasibility of this based on past work by your team and or additional information about the recruitment sites? I also do not understand why the sample size calculation is not grounded in your statistical approaches.

Dear editor, we wish to make a correction on that matter. The quantitative data collection started in March 2022 to February 2023 (IRB approval was in January 2022 as attached). The sample size calculation has been grounded in statistical approaches.

4. If you are interested in an outcome of depression, why are you not screening participants to enroll those with clinically significant depressive symptoms? There is no guarantee that you will result in a sample of MSM with HIV who also have depression that is large enough to facilitate your analysis

Depending on the research question and study design, it is more appropriate to include all HIV MSM regardless of their level of depressive symptoms. For example, if the goal of the study is to investigate the prevalence of depression among MSM living with HIV and its impact on HIV-related health outcomes, it would be important to enroll a representative sample of MSM living with HIV with varying levels of depressive symptoms. This would allow for prevalence to be measured and a more comprehensive understanding of the relationship between depression and HIV-related health outcomes in this population. However, it is debatable, and we fully respect your insight.

5. Inclusion criteria – “Men identified as homosexual based on clinic database”. This will exclude MSM who are not homosexual. MSM is a behavior, not an identity.

We have corrected the sentence. It would not impact the samples since the HIV databases in the respective clinics were based on self-identified men who have sex with men.

6. It seems that site selection is conflated with participant selection. For example – criterion 4 is that there must be more than 200 MSM with HIV. This isn’t a participant level factor.

Dear editor, criterion 4 has been removed, we only stated in the method section, that study location will be in the top ten clinics with the highest number of MSM living with HIV. This action also not affecting the sampling and sample size as the study location is still the same.

7. No where does it say that the participants must have HIV in the inclusion criteria

Thank you for the important point. We have added the criteria.

8. Section 4 social support. PLHIV is used and not defined

Thank you. All the abbreviations have been defined.

9. Might you consider measuring other likely relevant constructs including: self-efficacy in adhering to medications, internalized MSM related stigma

Thank you for the suggestion. For internalized stigma part of it has been grounded in the subscale of the 12-item short stigma scale. For self-efficacy and adherence to medication, we will consider it for the next research.

10. It is not clear how random sampling will occur.

In the quantitative arm of the study, proportionate random sampling was applied. The HIV databases from each clinic were obtained from the Selangor State Health Department to look for the list and the total number of patients in each clinic. Then, the required number of respondents in each clinic was calculated based on proportion to get 941 respondents. Based on the screened name list of MSM in each clinic, random sampling is conducted to select the required number of respondents using a randomizer application.

11. It is not clear why you would need to do a pilot study with 30 people to look for internal consistency of the select items. Have these measures not performed well in other samples? I could see you wanting to pilot if you were creating original measures.

Thank you. We totally agree with the statement. We have decided to remove the internal consistency conduct and only stick with the face validity and content validity.

12. The section on the ethical reasons of linking people who screen with depression to care should be included prior to the section on analysis.

Thank you. We agreed and followed the suggestion.

13. You have not written about exploring the nature of the missing data nor considered imputation to address missingness, which you could do with a sample size as large as you have.

We have updated the paragraph on handling missing data, followed the suggestion about exploring the nature of the missing data, and considered imputation to address missingness.

14. With a proposed sample size as large as you have and for a research question like you pose, it is a missed opportunity to not engage in structural equation modeling. You’re really interested in a lot of latent variables that are challenging to conceptualize and measure that affect the experience of depression in an individual who is MSM and also HIV+

Thank you for the suggestion. We will consider this in the next phase of the study.

15. Despite this being a mixed methods study, no attention is paid to the integration of the data.

Thank you for the very important input. We have explained the integration in the method and discussion section. The integration occurs in all research stages: study design, method, and report writing.

Discussion

1. I find it problematic to describe recall measures as buffering against the limitation of cross-sectional data. The data is still cross-sectional in nature. Perhaps instead of stating this, you could reiterate the value of the cross-sectional data (point prevalence of depression among a large sample of MSM is very useful, understanding of factors associated with depression informs intervention, or identifying factors that are protective against depression is useful too!)

Dear editor we totally agrred and have corrected the paragraph as suggested.

2. Overall – it is less important to be discussing how withdrawal of participants will be handled. I would save that for unpublished material. What would be most informative is a discussion of how this data will be integrated (quant/qual mixing), reiteration of the relevance of this work. ETC.

Thank you. We have removed the withdrawal part. We also have put a discsussion on the integration.

Dear Editors

Thank you again for your time and careful consideration of our manuscript. We appreciate the opportunity to respond to the reviewers' comments and hope we have addressed your concerns satisfactorily. We remain confident in the importance and validity of our findings, and we look forward to the possibility of publishing our work in your esteemed journal.

Sincerely

Zul Aizat Mohamad Fisal

Doctor of Public Health candidate,

Faculty of Medicine & Health Sciences,

Universiti Putra Malaysia 

On behalf of all authors.

---

## [Editor Report · Decision Letter 2]

4 May 2023

PONE-D-22-23739R2Biopsychosocial approach to understanding predictors of depressive symptoms among men who have sex with men living with HIV in Selangor, Malaysia: A mixed methods study protocolPLOS ONE

Dear Dr. Abdul Manaf,

Thank you for submitting your manuscript to PLOS ONE. After careful consideration, we feel that it has merit but does not fully meet PLOS ONE’s publication criteria as it currently stands. Therefore, we invite you to submit a revised version of the manuscript that addresses the points raised during the review process. Please see the attached file for my minor comments. 

We look forward to receiving your revised manuscript.

Kind regards,

Rogie Royce Carandang, RPh, MPH, MSc, PhD

Academic Editor

PLOS ONE
---

## [Author Response · Author response to Decision Letter 2]

8 May 2023

Dear Academic Editor and reviewers,

Thank you again for your comments and recommendations, which allowed us to improve the paper. We have tried to improve all the issues identified in the updated manuscript. In this document, we answer all the questions the reviewers highlighted.

Reviewers comments are shown in bold font, followed by our answer/comment in normal font. The corrections/changes in the manuscript are displayed through the tracked changes.

Comment and answer

Please add line numbers for easy review.

Line has been added

Abstract

Conclusion: toward Ending AIDS in 2030 � toward ending the AIDS epidemic by 2030

We have corrected the sentences throughout the manuscript.

Introduction

1. Depression symptoms � Depressive symptoms

Please use depressive symptoms throughout the text.

We have corrected the sentences throughout the manuscript.

2. ending AIDS 2023 � ending the AIDS epidemic by 2030

We have corrected the sentences throughout the manuscript.

3. Hypothesis: Your outcome variable is depressive symptoms. To present association, it

is better to show the direction from exposure to the outcome, not the other way

around. A (exposure) � B (outcome).

There is a significant association between depressive symptoms among MSM living

with HIV and social factors � There is a significant association between social

factors and depressive symptoms among MSM living with HIV.

We have corrected the sentences for the hypotheses.

Methods

1. Specific objective 3: Please reverse accordingly. Same comment as above.

We have corrected the sentences

2. Please improve the quality of figure 1. Include footnotes for abbreviations.

The figure has been improved to make it self-explanatory, abbreviations have been added, and figure has been converted using PACE.

3. Avoid using the term “respondents” once you have already identified your

respondents. It is better to specify than using general terms. Please update throughout

the text.

The “respondents” term has been changed to “MSM” as it is commonly used in academic research related to sexual health and behavior among men who have sex with men.

4. Page 15: for further management � for further assessment

Results from PHQ-9 will require clinical assessment.

The sentence has been changed as suggested.

5. Page 16: Do you have a tentative semi-structured topic guide? It will be helpful if you

will include this as a supplementary file.

Sure. The interview protocol was attached as a supplementary file.

6. Page 18: Please justify and cite your reference for choosing p-value <0.25 as criterion for the choice of variables to be included in multiple logistic regression model.

The reason has been justified with the sentences: A less rigorous threshold, such as a P-value < 0.25, should be applied to prevent omitting crucial variables from a model due to stochastic variation [38,39]. Therefore, variables with a P value < 0.25 will be included in multiple logistic regression to measure the predictors of depressive symptoms.

7. Page 24: Safety consideration � Safety considerations

We followed the suggestion.

References

1. Please follow PLOS ONES’ citation style, both in-text and reference list.

The citation has been adjusted to follow the Vancouver and PLOS ONES’ style. Duplicate citation were removed. 

Dear Editors

Thank you again for your time and careful consideration of our manuscript. We appreciate the opportunity to respond to your comments and hope we have addressed your concerns satisfactorily. If any problems arise, we are ready to improve the manuscript again. We remain confident in the importance and validity of our findings, and we look forward to the possibility of publishing our work in your esteemed journal.

Sincerely

Zul Aizat Mohamad Fisal

Doctor of Public Health candidate,

Faculty of Medicine & Health Sciences,

Universiti Putra Malaysia 

On behalf of all authors.

---

## [Editor Report · Decision Letter 3]

24 May 2023

Biopsychosocial approach to understanding predictors of depressive symptoms among men who have sex with men living with HIV in Selangor, Malaysia: A mixed methods study protocol

PONE-D-22-23739R3

Dear Dr. Abdul Manaf,

We’re pleased to inform you that your manuscript has been judged scientifically suitable for publication and will be formally accepted for publication once it meets all outstanding technical requirements.

Kind regards,

Rogie Royce Carandang, RPh, MPH, MSc, PhD

Academic Editor

PLOS ONE

---

## [Editor Report · Acceptance letter]

26 May 2023

PONE-D-22-23739R3 

Biopsychosocial approach to understanding predictors of depressive symptoms among men who have sex with men living with HIV in Selangor, Malaysia: A mixed methods study protocol 

Dear Dr. Abdul Manaf:

I'm pleased to inform you that your manuscript has been deemed suitable for publication in PLOS ONE. Congratulations! Your manuscript is now with our production department. 

Kind regards, 

on behalf of

Dr. Rogie Royce Carandang 

Academic Editor

PLOS ONE